# Diverse Polarimetric Features of AGN Jets from Various Viewing Angles: Towards a Unified View

Yuh Tsunetoe [1,*], Shin Mineshige [1], Tomohisa Kawashima [2], Ken Ohsuga [3], Kazunori Akiyama [4,5,6] and Hiroyuki R. Takahashi [7]

1 Department of Astronomy, Kyoto University, Kyoto 606-8501, Japan
2 Institute for Cosmic Ray Research, University of Tokyo, Tokyo 113-8654, Japan
3 Center for Computational Sciences, University of Tsukuba, Tsukuba 305-8577, Japan
4 Haystack Observatory, Massachusetts Institute of Technology, 99 Millstone Road, Westford, MA 01886, USA
5 Black Hole Initiative, Harvard University, 20 Garden Street, Cambridge, MA 02138, USA
6 National Astronomical Observatory of Japan, 2-21-1 Osawa, Mitaka-shi 181-8588, Japan
7 Department of Natural Sciences, Faculty of Arts and Sciences, Komazawa University, Tokyo 154-8525, Japan
* Correspondence: tsunetoe@kusastro.kyoto-u.ac.jp

**Abstract:** Here, we demonstrate that polarization properties show a wide diversity depending on viewing angles. To simulate images of a supermassive black hole and surrounding plasma, we performed a full-polarimetric general relativistic radiative transfer based on three-dimensional general relativistic magnetohydrodynamics models with moderate magnetic strengths. Under an assumption of a hot-jet and cold-disk in the electron temperature prescription, we confirmed a typical scenario where polarized synchrotron emissions from the funnel jet experience Faraday rotation and conversion in the equatorial disk. Further, we found that linear polarization vectors are inevitably depolarized for edge-on-like observers, whereas a portion of vectors survive and reach the observers in face-on-like cases. We also found that circular polarization components have persistent signs in the face-on cases, and changing signs in the edge-on cases. It is confirmed that these features are smoothly connected via intermediate viewing-angle cases. These results are due to Faraday rotation/conversion for different viewing angles, and suggest that a combination of linear and circular polarimetry can give a constraint on the inclination between the observer and black hole's (and/or disk's) rotating-axis and plasma properties in the jet–disk structure. These can also lead to a more statistical and unified interpretation for a diversity of emissions from active galactic nuclei.

**Keywords:** black hole physics; accretion disks; active galactic nuclei; radio jets; radiative transfer; polarimetry

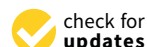



## 1. Introduction

The direct images of supermassive black holes (SMBH) M87* and Sgr A* by the Event Horizon Telescope (EHT) have opened a new era of black hole studies [1,2]. In particular, the polarimetric images around the black holes have attracted attention as they can reflect the configuration of magnetic fields around the SMBH [3]. It has been established from theoretical approaches that the magnetic fields should have an important role in the creation and acceleration of the jets from active galactic nuclei (AGNs) hosting the SMBHs [4,5]. We can thus expect to shed new light on the long-standing question of the driving mechanism of AGN jets through these unprecedented high-resolution observations.

However, one should also note that the polarized synchrotron emissions from a SMBH can experience Faraday effects on the way to Earth. Observational studies have detected traces of significant Faraday rotation of the linear polarization (LP) vectors in a range of millimeter and submillimeter wavelengths for many AGNs with/without jets [6–9]. It has been pointed out by theoretical studies that these should be attributed to internal Faraday rotation; that is, the Faraday rotation should occur in tandem with the emission

near an SMBH, e.g., [10–12]. Furthermore, recent calculations have demonstrated that significant Faraday conversion from LP to circular polarization (CP) can occur near a SMBH [11,13,14]. In such cases, CP components of the combination between the intrinsic emission and conversion from LP components can be a good tool for investigating the magnetic field structure.

In this way, it has been established that the unified interpretation of both LP and CP is essential for understanding the magnetic structure and other plasma properties around an SMBH. In this context, we have thought of calculating the polarization images theoretically. To predict images around and near a BH, we must take general relativistic (GR) effects into account, and, here, perform the GR radiative transfer (GRRT) calculation based on the GR magnetohydrodynamics (GRMHD) models, e.g., [15–17].

In the previous work Tsunetoe et al. [18], we confirmed, on the basis of our moderately magnetized models with a hot jet and cold disk, a scenario where the polarized emissions produced in the jet experience the Faraday rotation and conversion effects in the disk. In this description of the emitting jet and Faraday-thick disk, we found that the LP (or CP) intensities are mainly distributed in the downstream (upstream) side of the approaching jet for nearly face-on observers. In this work, we surveyed polarimetric features for different inclination angles between the black hole's spin-axis and an observer, to expand and develop the discussions. Here, we bore in mind observations of a diverse range of AGN jets at 230 GHz by the EHT and other very long baseline interferometers (VLBIs). Furthermore, we thought of applying them to the interpretation of the disk precession around SMBHs [19,20].

## 2. Methods

We followed the model parameters adopted in the fiducial model in [18]. Here, we used a three-dimensional GRMHD model simulated with `UWABAMI` code [21,22], which is categorized into the intermediary area between a magnetically arrested disk (MAD; [23]) and standard and normal evolution (SANE; [24]), and is thus called semi-MAD. The $R - \beta$ model was adopted for the determination of electron temperature, where the proton–electron temperature ratio ($T_i/T_e$) was given at each point in the fluid model by a function of plasma-$\beta$ ($\equiv p_{\rm gas}/p_{\rm mag}$; gas–magnetic-pressure ratio) and two parameters $R_{\rm low}$ and $R_{\rm high}$ [25] as follows:

$$\frac{T_i}{T_e} = R_{\rm low}\frac{1}{1+\beta^2} + R_{\rm high}\frac{\beta^2}{1+\beta^2}.$$ (1)

The model parameters are summarized in Table 1. GRRT calculation was performed by a code implemented in [11,18,26] with a sigma cutoff of $\sigma_{\rm cutoff} = 1$, and fast-light approximation was performed for a snapshot fluid model. Here, we used a snapshot at $t = 9000t_g$ (here, $t_g \equiv r_g/c$) in the quasi-steady state as a main model, while three other snapshots are surveyed and discussed in Section 4.2. The mass accretion rate onto the black hole was fixed so that we reproduced the 230 GHz observed flux of M87 in [1], $\approx$0.5 Jy, for the case with $i = 160°$. Under these assumptions, we calculated the total, LP, and CP images, varying the observer's inclination (viewing) angle $i = 0°–180°$ by $10°$ as pictured in Figure 1.

**Table 1.** Parameters in our GRMHD and GRRT model, following a fiducial model in [18], except the observer's inclination angle. Here, $M_\odot$ is the solar mass. $\phi \equiv \Phi_{\mathrm{BH}}/\sqrt{\dot{M}r_\mathrm{g}c^2}$ is a strength of dimensionless magnetic flux on the event horizon of SMBH, where $r_\mathrm{g} \equiv GM_\bullet/c^2$ and $\Phi_{\mathrm{BH}} = (1/2) \int \int |B^r| \mathrm{d}A_{\theta\phi}$. $G$ and $c$ are the gravitational constant and speed of light, respectively.

| Parameter | Value |
|---|---|
| BH mass $M_\bullet$ | $6.5 \times 10^9 M_\odot$ |
| BH spin parameter $a$ | 0.9375 |
| Magnetic flux on the horizon $\phi$ | $\approx 18$ (semi-MAD) |
| $T_\mathrm{e}$-parameter $R_{\mathrm{low}}$ | 1 |
| $T_\mathrm{e}$-parameter $R_{\mathrm{high}}$ | 73 |
| Mass accretion rate onto BH $\dot{M}$ | $6 \times 10^{-4} \, M_\odot \mathrm{yr}^{-1}$ (at the moment of snapshot) |
| Distance to observer $D$ | 16.7 Mpc |
| Observational frequency $\nu$ | 230 GHz |

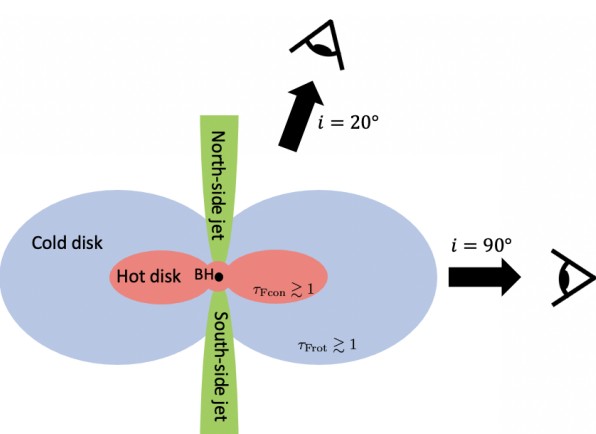

**Figure 1.** A schematic picture of the jet–disk structure (adopted from Tsunetoe et al. [18]). In our model, synchrotron emission is predominantly produced in the funnel jet (green). After the emission, the polarized lights experience Faraday conversion in the inner hot disk (red) and Faraday rotation in the outer cold disk (blue) on the way to the observer (eyeball), respectively. See [18] for poloidal-slice maps of typical values of the emissivity and Faraday coefficients.

## 3. Results

### 3.1. Polarization Images

In Figure 2, we show resultant polarization images at 230 GHz for three inclination angles of $i = 20°, 50°$, and $90°$ as examples of the nearly face-on, intermediate, and edge-on cases, respectively. The observers for the first and third cases are pictured by arrows and eyeballs in Figure 1. (A movie of all of the images for $i = 0°$–$180°$ can be found on https://youtu.be/065qAx6Tff0, accessed on 16 October 2022). The $i = 20°$ case in the top panels shows polarimetric features following the fiducial image in [18] for $i = 160°$; that is, the total intensity showing a photon ring and dim tail-like jet component overlapped on the ring, the LP map with partly scrambled vectors, and the CP image giving a bright, negative ring.

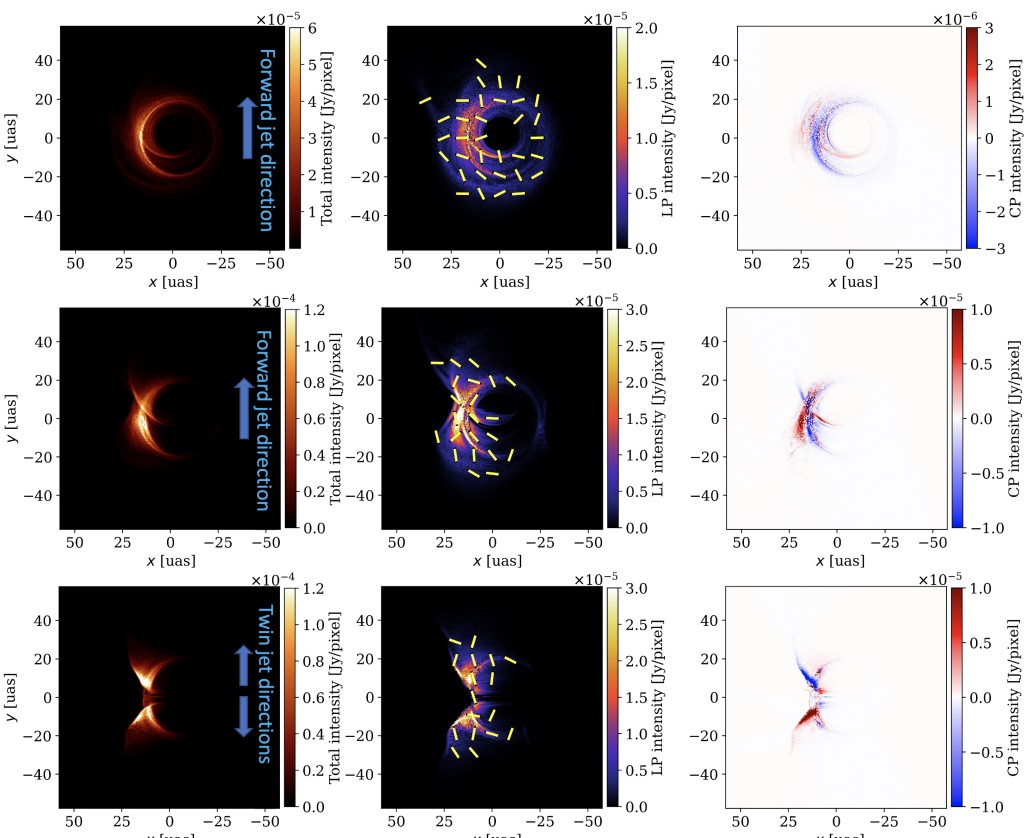

**Figure 2.** A calculated polarization images at 230 GHz for three viewing angles of $i = 20°$ (nearly face-on), $50°$ (intermediate), and $90°$ (edge-on), top to bottom. (**Left**) Total intensity (Stokes $I$) image. Each image consists of $600 \times 600$ pixels. The forward jet extends upward on the image. (**Center**) LP map. The LP intensity is shown by the color contour, with LP vectors in electric vector position angle (EVPA) overwritten. (**Right**) CP image. The CP intensity (Stokes $V$) is shown by the color contour with sign. A movie of all the images for $i = 0$–$180°$ can be found on https://youtu.be/065qAx6Tff0 (accessed on 16 October 2022).

Here, we see a reversal of the CP ring's sign compared to that for $i = 160°$ in [18] (which shows a positive ring), including the fine "second sub-ring" (see [14]) and the tail-like jet on the ring with positive signs (which were negative for $i = 160°$). These are attributed to the helicity of the magnetic fields, which is flipped between the northern and southern part about the equatorial plane in our model. This is because of the frame-dragging effect of a rotating BH (for detailed discussion on CP's sign, see also [14,26]). A similar sign-reversal of CP rings for two observers in the north and south side of the BH was also reported by [13].

The edge-on images in the bottom panels of Figure 2 exhibit distinct morphological features compared to the nearly face-on ones. The total intensity image shows a crescent-shaped BH shadow broken at the equator, which is caused by a synchrotron self-absorption (SSA) effect in the optically thick disk. This broken crescent is qualitatively different from a three-forked shadow in [26], who modeled Sgr A* with a hot disk, implying the effect of the disk temperature on the shape of the BH shadow.

The edge-on LP maps give a scrambled vector pattern (which will also be shown by a total LP fraction in the next subsection), whereas the CP image follows a total image with flipping signs in four parts of the image divided by the left-leaning (relativistically beamed) vertical line and the equatorial line. This CP sign-reversal in the four parts is due to the intrinsic emission and the Faraday-rotation-induced conversion with the reversal of the magnetic field helicity about the equatorial plane [14,26].

Further, we show the images for $i = 50°$ in middle panels of Figure 2 as an example of intermediate inclination cases. The images show intermediate features between the face-on case and edge-on case, giving a crescent-shaped shadow and tail-like emission from the approaching jet. In the CP image, we can also see both a blue, ring-like feature and a sign-flipping feature on the jet-edge.

### 3.2. Unresolved Polarimetric Features

For a more comprehensive survey of inclination angles, we next show a diagrams of the total (image-integrated) intensity flux, $I_{total} = \sum_{pixels} I$, in Figure 3. (Note that, here, we assume a distance to M87 of $D = 16.7$ Mpc for all cases.) The profile is steep for nearly face-on cases and flat for nearly edge-on cases, and has a symmetry of the total flux about the equatorial plane (the edge-on case) with two peaks at $i \approx 50°$ and $\approx 130°$.

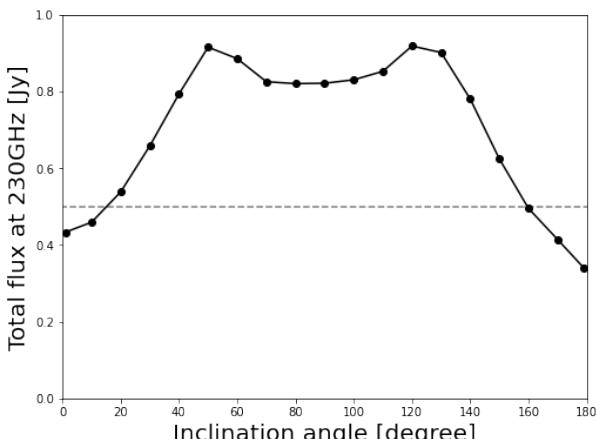

**Figure 3.** Diagram of the total (image-integrated) fluxes at 230 GHz for different inclination angles, assuming the distance to M87. Gray dash line corresponds to 0.5 Jy.

Next, a diagram of the total LP and CP fractions, $\sqrt{Q_{total}^2 + U_{total}^2}/I_{total}$ and $V_{total}/I_{total}$ (here, $(I, Q, U, V)$ are the Stokes parameters), is shown in Figure 4. Here, we find a roughly symmetric feature in the LP fractional profile with higher fractions in more face-on like cases. In contrast, the CP fractional profile shows an antisymmetric profile about the $i \approx 90°$ case with three peaks and bottoms.

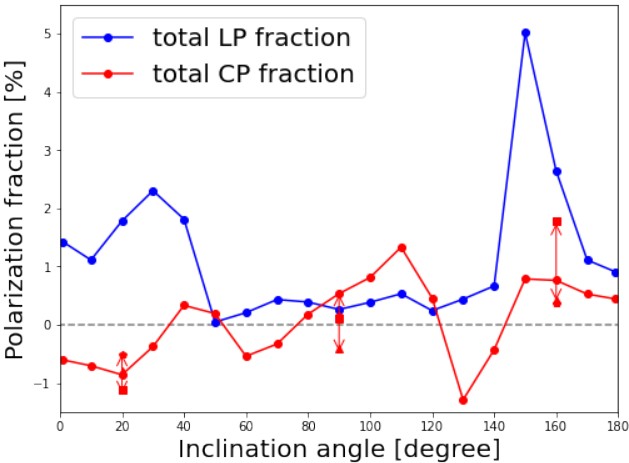

**Figure 4.** Diagram of the total LP and CP fractions at 230 GHz for different inclination angles. For $i = 20°, 90°$, and $160°$ cases, we additionally plotted the values for other three snapshots in the GRMHD model. $t = 9500t_g$ (triangle), $10,000t_g$ (square), and $11,000t_g$ (pentagon), whereas $t = 9000t_g$ (circle) for the fiducial model. The arrows indicate the time variations of the CP fractions.

## 4. Discussion

In this section, we will pick up representative polarimetric features that appear in the images and total (unresolved) values obtained in the last section, and discuss their relationship with the magnetic field configuration and plasma structure around the SMBH.

### 4.1. Total Flux Suppression for the Edge-On Like Cases

We saw a two-hump feature at $i \approx 50°$ and $\approx 130°$ in the total flux profiles in Figure 3. This can be attributed to the opening angle of the funnel jet region around the black hole, where the emissions are predominantly produced (see also Figure B1 in [18]). In the case where the observer's inclination is smaller than the opening angle of the jet, as $i \leq 50°$, $130° \leq i$, the emissions go through the sparse funnel region. Otherwise, as $60° \leq i \leq 120°$, the emissions go through the dense disk and experience a significant SSA effect in the disk.

In Figure 5, we show an inclination diagram of a typical (image-integrated and intensity-weighted) optical depths for Faraday rotation/conversion and SSA (see [18] for a definition and introduction of these optical depths). In fact, typical optical depths for the SSA (orange profile) approach $\sim 1$ at $i = 50°$ and $130°$, where the plasma transit between optically thin and thick states. As a result, the intensities are saturated in the foregrounded colder plasma and suppressed in the edge-on-like cases.

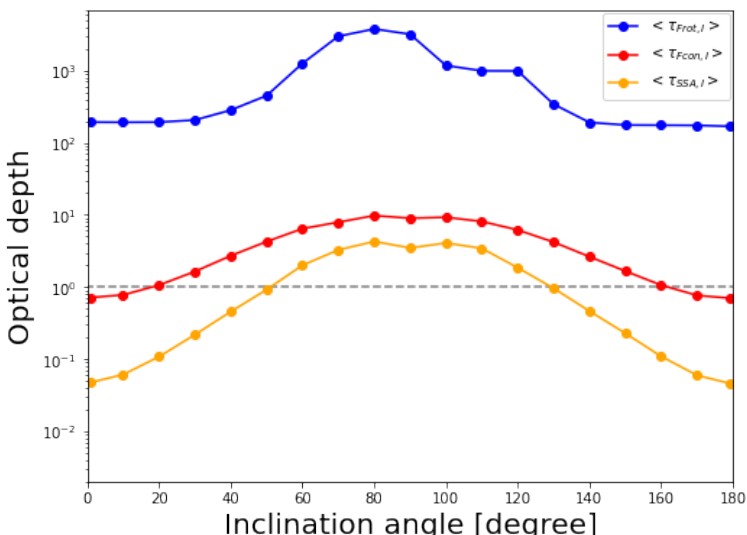

**Figure 5.** Diagram of the image-integrated intensity-weighted optical depths for Faraday rotation and conversion, and SSA at 230 GHz for different inclination angles. Gray dashed line corresponds to $\tau = 1$.

### 4.2. Reversal of Unresolved CP Signs

In Figure 4, we find a reversal of CP signs for face-on like cases in the north and south side of the SMBH; that is, $i \approx 0°$–$30°$ cases persistently give negative CP fractions whereas $i \approx 150°$–$180°$ show positive ones. These persistent CP signs result from integrating the CP images that consist of a monochromatic photon ring (e.g., the left panel in Figure 2), which are unique to the face-on-like cases and are due to the helicity of magnetic fields.

We additionally plot points of CP fractions for the other three snapshots (at $t = 9500t_g$, $10,000t_g$, and $11,000t_g$) for $i = 20°, 90°$, and $160°$ in Figure 4. They show persistent signs of time-variable CP fractions for two face-on-like cases, while their absolute values can vary by a factor. Meanwhile, the CP fractions for the edge-on case change their sign for time-variability and are relatively small in their absolute values. This is because CP intensities with both signs comparably contribute to the unresolved CP flux (see the bottom-right panel in Figure 1). Due to the cancellation of its sign, the CP fraction is small and the time-variable emission can easily change the sign of the total CP flux. These results suggest

that we can survey the magnetic field configuration around the SMBH through the circular polarimetry, even for unresolved sources.

From stimulated CP observations of AGNs, the following facts have been established; if some AGNs show significant positive (or negative) CP fluxes, they continue to show positive (negative) CP fluxes over timescale of decades. The same is true for those AGNs that do not show significant CPs [27]. For example, Sgr A* are known to continue to show negative CP fluxes at centimeter to submillimeter wavelengths [8,28]. In contrast, quasar 3C 279 always shows positive CPs as a whole [29,30].

We can interpret these observational results in relation to our calculated values for the face-on-like cases, which are consistent with the low inclination angle interpretation favored for these two objects [31–33]. However, we should note that the "core" emission in large beam-sized observations of low-inclination objects can degenerate the distant, downstream jet/outflow components, which would be resolved as "knots" at a higher resolution, with the emission from around the central SMBH.

*4.3. Symmetry of the LP-CP Separation along the Jet*

Here, we survey the separation features among the total, LP, and CP intensity distributions on the images. In [18], we demonstrated that the LP components (or CP components) are distributed in the downstream (or upstream/counter-side) of the approaching jet on the images for $i = 160°$ due to Faraday conversion in the inner hot disk and rotation in the outer cold disk as pictured in Figure 1.

We applied these analyses to the cases with different inclination angles. The separations between total and LP (or CP) intensities are shown in Figure 6, in which, the peak separations of the cross-correlation function between two kinds of intensities $I - P$ and $I - |V|$ are plotted (see [18] for detail). Here, we convolved the images with a circular Gaussian beam of 17 μas, bearing present and near-future EHT observations in mind.

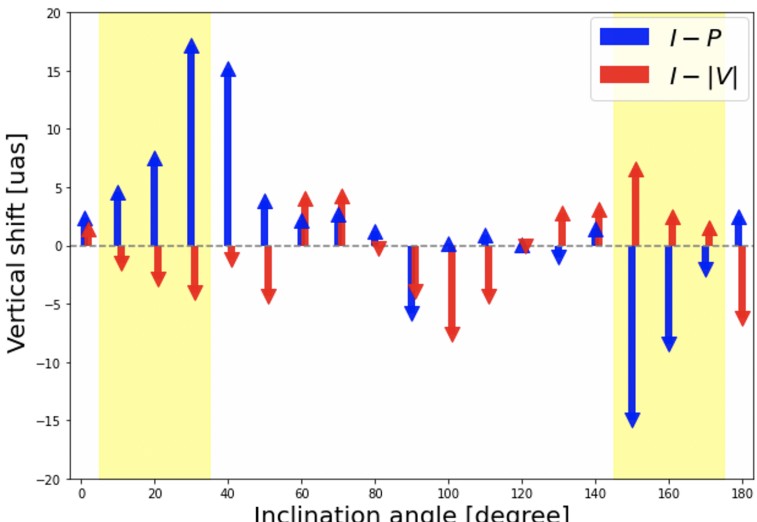

**Figure 6.** Diagram of separations in *y*-direction in Figure 2 (the direction of projected SMBH spin axis) between total and LP (or CP) intensities at 230 GHz for different inclination angles. The LP (or CP) intensities tend to be distributed downstream (upstream) of the jet for Faraday thick cases. Arrows point to the distances of separations in the vertical direction of the images between total and LP (or CP) intensity distribution in blue (or red) color. The separations are calculated from cross-correlation function between two kinds of intensities on the images. See Tsunetoe et al. [18] for introduction and definition of the correlation analyses.

We can clearly see the LP-CP separations along the jet in the face-on-like cases both from the north ($i = 10°$–$30°$) and south side ($i = 150°$–$180°$), which are yellow-marked, introduced at the beginning of this section. We showed in [18] that the larger the inclination angle, the large the LP-CP separations become due to the longer projected distance on the

screen, by using the latter three cases. Here, we also confirm this tendency in the northern face-on-like cases, where the directions of separation are reversed for the north-side cases because the approaching (north-side) jet extends upward on the images.

These results demonstrate that the description of the synchrotron-emitting jet and Faraday-thick disk, as pictured in [18] and Figure 1, can be applied to the face-on cases both in north and south sides. In fact, it is shown in Figure 5 that typical optical depths for Faraday rotation (blue) and conversion (red) have a symmetric structure for the face-on-like observers in the north and south sides.

### 4.4. Oscillation of CP Signs

Next, we examined an oscillation feature of CP signs in edge-on-like cases. In Figure 4, we see a hint of flipping CP signs that oscillate for two cycles in a range of $i \approx 40°$–$140°$. This can be interpreted with a combination of polarimetric features introduced so far. In the following descriptions, we distinguish two cases, observed from the north side and from the south side. Figure 2 shows the former cases (i.e., $i \leq 90°$).

First, the CP images in edge-on-like cases are characterized by changing signs between the neighboring quadrants, as seen in the bottom-right panel in Figure 2. We confirm the validity of this description for edge-on "like" (not only $i = 90°$) cases.[1] In particular, the CP intensities in the **second and third quadrants** are brighter due to the relativistic beaming effect (see the bottom-right of Figure 2), and are dominant for the image-integrated, unresolved CP flux. Second, the unresolved CP fluxes in lower inclination cases are predominantly contributed from the counter-side jet (as shown in Section 4.3); the stronger CP intensities are found in the **third quadrant** (or **second quadrant**) side for the north (south) side observers. Third, the SSA effect becomes significant for a large inclination angle, as shown in Figure 5. Then, the CP emissions from the counter-side jet are suppressed because of large optical depths, and yield their dominance to those from the foreground-side jet (**second quadrant** for north and **third quadrant** for south). This is also shown by upward and downward $I - |V|$ arrows for north and south side cases in Figure 6, respectively.

As a result, the dominant part for the CPs changes in order of the negative ring (face-on cases in the north), **third quadrant** (positive), **second quadrant** (negative), ($i = 90°$; crossing the equatorial plane,) **third quadrant** (positive), **second quadrant** (negative), and the positive ring (face-on in the south), if starting from $i = 0°$ to $180°$. In this way, the oscillation of the total CPs is explained with the monochromatic rings and quadranted images.

Ricarte et al. [14], using MAD and SANE models, also calculated profiles of unresolved CP fractions for inclination angles. Their profiles give negative and positive values for face-on-like cases in the north and south side, respectively, in a similar way to ours. Meanwhile, they show a sign-changing feature for edge-on cases but for one cycle (cf. two cycles in our case). The difference may be explained by removing our third sign-changing factor above: the change in the bright region. Where the emission, Faraday rotation/conversion, and SSA occur depends on many factors, such as the magnetic strength and electron temperature and density. Thus, the difference in the MAD-SANE regime and the electron temperature prescription can drastically affect the morphology of the images and integrated CP fractions.

The total CP fraction is a product from integrating an image consisting of the intrinsic emission and rotation- and twisted-field-driven conversion, which have different dependencies on the plasma and observational properties to each other. Thus, it may be difficult to access the characteristics of the system through this unresolved quantity alone [14]. One straightforward application is to combine it with the total LP, which we will discuss in the next subsection.

### 4.5. Combination of the Unresolved LP and CP Fractions

Finally, we focus on the relationship between the unresolved LP and CP fractions. The unresolved LP fractions in Figure 4 give a symmetric-like profile with high ($\gtrsim 1\%$) and low ($\lesssim 1\%$) values in face-on and edge-on-like cases, respectively. This result is due

to a larger optical depth for Faraday rotation for larger inclination angles. As pictured in Figure 1, the emitted LP vectors to the more edge-on-like observer experience a larger Faraday rotation in the outer cold disk. In fact, the typical optical depths for Faraday rotation are larger in the edge-on cases by approximately one order of magnitude than in the face-on cases.

If we combine this with the discussion in Section 4.2, we can conclude that the unresolved LP and CP fractions are characterized by relatively strong LPs and sign-persistent CPs in the face-on-like cases, and weak LPs and time-variable CPs in the edge-on cases. Precisely speaking, the CP signs are time-varying in the edge-on-like cases (see Figure 4).

M87(*) has been known to show strong LP and weak CP flux at radio wavelenths [2,6,34]. If we apply the model constraints for the total LP and CP fractions from Event Horizon Telescope Collaboration [35], the face-on like models are favored, which is consistent with the well-known large-scaled M87 jet. Future stimulating resolved/unresolved LPs and CPs will become a good tool for investigating the system of SMBH and plasma in M87 itself, and for applying knowledge of M87 to other AGN jets.

In contrast, M81* has been reported to show larger CP fractions than LP fractions [36,37]. These observation may be explained by our edge-on-like $i \approx 40°–140°$ cases with low LPs due to strong Faraday rotation, which is also consistent with high inclination angles referred for radio galaxies.

Whether and how source types, such as blazars, quasars, and radio galaxies, are related to the LP and CP fractions [29,38] has also been discussed. Coupled with more statistical data from the future resolved/unresolved spectro-polarimetry of various targets, a survey of inclination angles can give a clue for accessing a unified description of a diversity of AGNs.

### 4.6. Future Prospects

In this work, we focused our discussion on the diverse appearance of AGN jets, using the same fluid model, to demonstrate how jets are observed differently depending on the viewing angle. Meanwhile, it should be surveyed whether the results based on the semi-MAD model are common even for other fluid models, such as more SANE- or MAD-like models, and/or with different electron-temperature prescriptions, including the time-variability. More statistical surveys for various fluid models and long time durations is the scope of our future works.

**Author Contributions:** Investigation, Y.T.; Supervision, S.M.; Writing—original draft, Y.T.; Writing—review & editing, S.M., T.K., K.O., K.A. and H.R.T. All authors have read and agreed to the published version of the manuscript.

**Funding:** This work was supported in part by JSPS Grant-in-Aid for JSPS fellows JP20J22986 (Y.T.), for Early-Career Scientists JP18K13594 (T.K.), for Scientific Research (A) JP21H04488 (K.O.), and for Scientific Research (C) JP20K04026 (S.M.), JP18K03710 (K.O.), and 20H00156, JP20K11851, JP20H01941 (H.R.T.). This work was also supported by MEXT as "Program for Promoting Researches on the Supercomputer Fugaku" (Toward a unified view of the universe: from large scale structures to planets, JPMXP1020200109) (K.O., T.K. and H.R.T.), and by Joint Institute for Computational Fundamental Science (JICFuS, K.O.). KA is financially supported in part by grants from the National Science Foundation (AST-1440254, AST-1614868, AST-2034306). Numerical computations were in part carried out on Cray XC50 at Center for Computational Astrophysics, National Astronomical Observatory of Japan.

**Data Availability Statement:** The data that support the findings of this study are available from the corresponding author, Y.T., upon reasonable request.

**Conflicts of Interest:** The authors declare no conflict of interest.

## Abbreviations

The following abbreviations are used in this manuscript:

| | |
|---|---|
| SMBH | SuperMassive Black Hole |
| EHT | Event Horizon Telescope |
| AGN | Active Galactic Nucleus |
| LP | Linear Polarization |
| CP | Circular Polarization |
| GRRT | General Relativistic Radiative Transfer |
| GRMHD | General Relativistic MagnetoHydroDynamics |
| SANE | Standard And Normal Evolution |
| MAD | Magnetically Arrested Disk |
| SSA | Synchrotron Self-Absorption |

## Note

[1]　Refer to the movie of all of the images for $i = 0°$–$180°$ on https://youtu.be/065qAx6Tff0 (accessed on 16 October 2022).

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
