# Peer review of "Diverse Polarimetric Features of AGN Jets from Various Viewing Angles: Towards a Unified View"

_galaxies, doi:10.3390/galaxies10050103_

Round 1

Reviewer 1 Report

This manuscript has investigated the viewing angle dependence on the polarization properties of black hole shadow images. They found that clear dependency on the viewing angle for linear and circular polarizations.  It is interesting research and relevant for the topics of this special issue. Therefore, I recommend that this manuscript should be published in Galaxies.

But this manuscript is still incomplete. So the authors should address following issue.

Major comments:

1.

In Sec. 2, the authors used GRMHD simulation data as an input data. It is still unclear what GRMHD simulation data the authors used. Are GRMHD simulations 2D or 3D? What simulation time do the authors use for reading? Do the authors use only snapshot data or sequential simulation data? I think the authors fixed mass accretion rate for obtained GRRT image. What is the total flux at 230GHz?
It is worth to provide more information about how to adopt GRMHD simulations for GRRT calculation.  

2.

All results are based on snapshot images. I wonder these results are general trend in different simulation time. I think it is better to show time-average value and deviation for different simulation time (snapshots).

3.

I think the results obtained the authors depend on R_high value. Have you check different R_high value cases? I think it is worth to mention about this point.

Minor comments:

4.

In Sec. 3.1, first sentence, “… for two inclination angles …” => “… for three inclination angles …”

5.

In P.6, first sentence, “… the time-varible emission …” => “… the time-variable emission ….”

Reviewer 2 Report

The article is dealing with very relevant subject (black hole image polarization), is reasonably written, so I must only recommend it for publication.

Author Response

We wish to thank the reviewer for reading and helpful comments. 

Round 2

Reviewer 1 Report

The authors have addressed all issues and well-revised manuscript. I do not have any further comments. So, I recommend that this manuscript for the publication at the present form.